

# Neutrophil–lymphocyte ratio is prognostic in early stage resected small-cell lung cancer

Zoltan Lohinai[1,2], Laura Bonanno[3], Aleksei Aksarin[4], Alberto Pavan[3], Zsolt Megyesfalvi[1,2], Balazs Santa[1], Virag Hollosi[1], Balazs Hegedus[5], Judit Moldvay[1], PierFranco Conte[6], Mikhail Ter-Ovanesov[7], Evgeniy Bilan[8], Balazs Dome[1,2,9] and Glen J. Weiss[10]

[1] National Koranyi Institute of Pulmonology, Budapest, Hungary
[2] Department of Thoracic Surgery, Semmelweis University and National Institute of Oncology, Budapest, Hungary
[3] Medical Oncology 2, Istituto Oncologico Veneto IOV IRCCS, Padova, Italy
[4] Surgut District Clinical Hospital, Surgut, Russia
[5] Department of Thoracic Surgery, University Hospital Essen, Essen, Germany
[6] Department of Surgical, Oncological and Gastroenterological Sciences, Università degli Studi di Padova, Padova, Italy
[7] Peoples' Friendship University of Russia, Moscow, Russia
[8] Department of Oncology, Surgut District Clinical Hospital, Surgut, Russia
[9] Division of Thoracic Surgery, Department of Surgery, Medical University of Vienna, Vienna, Austria
[10] Beth Israel Deaconess Medical Center/Harvard Medical School, Boston, MA, USA

Corresponding author
Zoltan Lohinai,
zoltan.lohinai@koranyi.hu

## ABSTRACT

**Background:** For selected early stage small cell lung cancer (SCLC), curative intent surgery is often performed. Previous studies, predominantly from East Asia, reported that high neutrophil to lymphocyte ratio (NLR), and platelet–lymphocyte ratio (PLR) correlate with poor prognosis in several types of tumors including SCLC. Our aim was to investigate the prognostic value of NLR and PLR in Caucasian patients with resected SCLC, as potential tool to select patients for multimodal treatment including surgery.

**Methods:** Consecutive patients evaluated at three centers between 2000 and 2013 with histologically confirmed and surgically resected SCLC were retrospectively analyzed. NLR and PLR at diagnosis was used to categorize patients into "high" and "low" groups based on receiver operating curve analysis. Univariate and multivariate analyses were used to evaluate the impact of clinical and pathological characteristics on outcome.

**Results:** There were a total of 189 patients with a median age of 58 years, and the majority had stage I or II disease. We found a significant correlation between NLR and tumor stage ($p = 0.007$) and age ($p = 0.038$). Low NLR (LNLR) was associated with significantly longer overall survival, while PLR had no prognostic impact. There were significant associations between NLR and PLR but not with gender, vascular involvement, tumor necrosis, peritumoral inflammation, or tumor grade.

**Conclusion:** Pre-operative LNLR may be a favorable prognostic factor in stage I–II SCLCs. PLR is not prognostic in this population. LNLR is easy to assess and can be integrated into routine clinical practice. Further prospective studies are needed to confirm these observations.

## INTRODUCTION

Small cell lung cancer (SCLC) is a highly aggressive malignancy with a poor prognosis (*Siegel, Miller & Jemal, 2017*). In most cases it is diagnosed in advanced-stages, therefore surgical resection is rarely possible and serves little clinical benefit (*Lad et al., 1994*). In recent years, for selected early stage cases, curative intent surgery has been offered, however there remains no validated predictive or prognostic biomarker for these patients. Two large cohort study suggests an increased role of surgery in multimodality therapy for early stage SCLC (*Combs et al., 2015*; *Yang et al., 2017*). These results might indicate that surgery could be a treatment option even in locally-advanced disease after appropriate patient selection.

Since the late 1990s, evidence has been available concerning the role of tumor infiltrating immune system cells in tumor progression (*Schäfer & Werner, 2008*). Routine blood tests based on the neutrophil to lymphocyte ratio (NLR) and platelet to lymphocyte ratio (PLR) are potential biomarkers of the systemic inflammatory response and might serve as prognostic biomarkers for survival across a variety of malignancies (*Guthrie et al., 2013*; *Gu et al., 2016*). Previous studies predominantly from East Asia have shown that high PLR (HPLR) and high NLR (HNLR) correlate with poor prognosis in several tumor types (*Zhou et al., 2014*; *Proctor et al., 2012*).

In non-small cell lung cancer (NSCLC) there is data showing that NLR and PLR can be prognostic in advanced and early stage cohorts including surgically resected patients (*Unal et al., 2013*; *Takahashi et al., 2016*). However, in SCLC, NLR, and PLR studies were reported exclusively from East Asia, and data have not yet been published on Caucasian or resected stage populations (*Siegel, Miller & Jemal, 2017*; *Lad et al., 1994*; *Combs et al., 2015*; *Yang et al., 2017*; *Schäfer & Werner, 2008*; *Guthrie et al., 2013*).

All in all, although surgical treatment of SCLC is challenging, in this study, we aim to identify biomarkers that help in patient selection that potentially benefit from lung resection.

We hypothesized that potential indicators of the inflammation such as PLR and NLR is associated with SCLC prognosis. Therefore, to our knowledge, this is the first Caucasian study that evaluates the association of NLR and PLR with the prognosis in surgically resected, limited stage SCLC patients.

## MATERIALS AND METHODS

### Ethics statement

The study was conducted in accordance with the guidelines of the Helsinki Declaration of the World Medical Association and with the approval of the national level ethics committee (Hungarian Scientific and Research Ethics Committee of the Medical Research Council, ETTTUKEB-7214-1/2016/EKU), which waived the need for individual informed consent for this retrospective study. The Ethics Committee of Istituto Oncologico

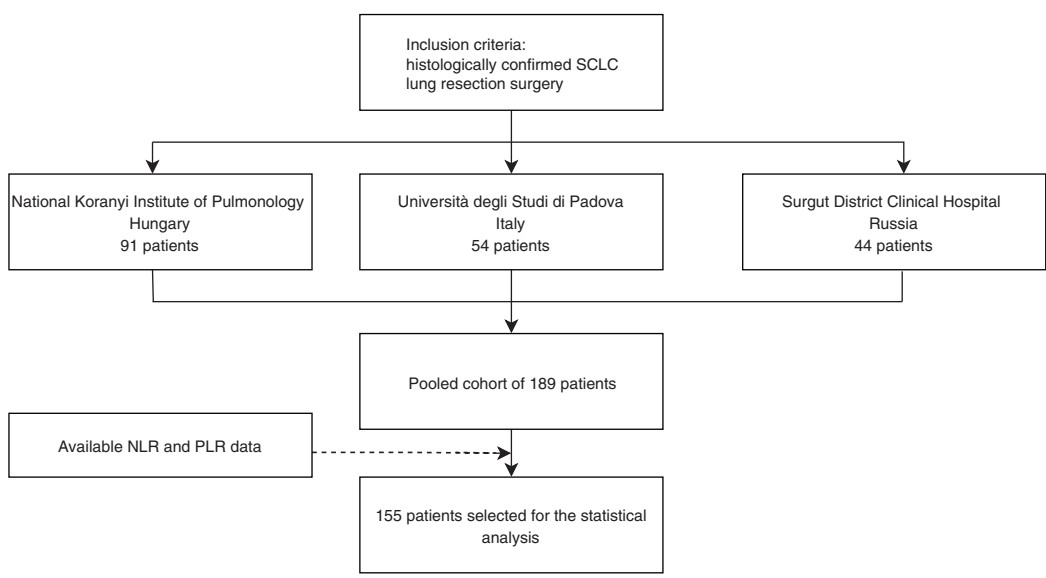

**Figure 1 Study design of surgically resected SCLC patient's cohort ($n = 189$) according to institution and PLR and NLR data availability.**

Veneto in Padova (Italy) evaluated and approved the study and patients provided informed consent to be signed for the collection, analysis and publication of data, when the investigators have the possibility to collect it from patients, according to Italian data protection authority dispositions. The ethics Committee of the Department of Health in Ugra (Russia) has evaluated and approved the study, which waived the need for individual informed consent for this retrospective study. Clinical information was collected and patients were de-identified. Subsequently, patients cannot be identified either directly or indirectly.

## Study population

In this retrospective study, surgically resected and histologically confirmed consecutive SCLC patients evaluated at three centers, between 2000 and 2013 at the National Koranyi Institute of Pulmonology (cohort #1), between 2000 and 2013 at Università degli Studi di Padova (cohort #2), and between 1999 and 2013 at the Surgut District Clinical Hospital (cohort #3) were included (Fig. 1).

Data from all three institutions were measured and graded with comparable ways across the different institutions.

Histologic diagnosis was performed according to the same contemporary pathology guidelines across all three centers. Clinicopathological data collected included gender, age, smoking history, chemo- (CHT) and radiation therapy (RT) treatments, type of operation (segmentectomy, lobectomy, and pneumonectomy), and overall survival (OS) and/or disease free survival (DFS). TNM stage according to the Union for International Cancer Control (7th edition) (*Mirsadraee et al., 2012*), and patient age at the time of diagnosis was recorded. OS was estimated from the time of diagnosis, until death, or the last available follow-up visit. The study and all treatments were conducted based on the individual

institutional guidelines in accordance with the current National Comprehensive Cancer Network guidelines with no major differences across the three centers.

Cut-off for the date of the last follow-up visit included in this analysis was set to April 2017. Blood samples were obtained from routine hematologic control using the same reference values across the three centers. Most studies examining NLR and PLR did not explicitly state the time frame of blood sample analysis prior to surgery, thus we set the time point to be within 30 days prior to surgery, so that the data could be broadly applicable to clinical practice. The NLR and PLR were calculated according to the absolute neutrophil and platelet counts and absolute lymphocyte counts on routine blood tests obtained within 30 days prior to surgery. NLR values were obtained by dividing the absolute neutrophil count by the absolute lymphocyte count, while PLR values were obtained by dividing the absolute platelet count by the absolute lymphocyte count. For patients who received neoadjuvant chemotherapy, blood tests prior to initiation of neoadjuvant chemotherapy were analyzed.

Based on the relatively low case numbers in each individual patients' cohort and the lack of significant differences in OS and DFS according to cohorts ($p = 0.360$ and $p = 0.744$, respectively), we pooled cohorts #1–3 together. There were no significant differences in gender or stage across the three individual cohorts, however, there were significant differences in age, CHT and RT administration (Tables S1 and S2).

## Treatment

Patients who underwent complete tumor resection including segmentectomy, lobectomy, and pneumonectomy with mediastinal lymph node dissection were included in this retrospective analysis. Selected patients were treated with postoperative systemic CHT with a platinum-etoposide doublet regimen or with a combination of cyclophosphamide, epirubicin, and vincristine. Selected patients with positive mediastinal lymph node status received preoperative CHT with the aforementioned regimens.

## Statistical methods

Statistical analyses were performed only on patients with available PLR and NLR data ($n = 155$). Since there were no optimal cut-off values reported in resected limited-stage SCLC for NLR and PLR, time-dependent receiver operating curve (ROC) analysis was used to identify the optimal cut-off values for these parameters according to OS and DFS. ROC was applied to determine the optimal cut-off, as the values whose sensitivity and specificity are the closest to the value of the area under the ROC curve and the absolute value of the difference between the sensitivity and specificity values is minimal, as described in other studies (*Coffelt et al., 2010*; *DeNardo, Andreu & Coussens, 2010*). Next, patients were divided into "high" and "low" groups according to their preoperative NLR and PLR.

Kaplan–Meier curves and two-sided log-rank tests were used for univariate survival analysis. Two-sided $p$-values less than 0.05 were considered statistically significant. All variables with $p$-values less than 0.05 were included in the multivariate analysis. The Cox proportional hazards model was used for univariate and multivariate survival

analyses to calculate the hazard ratios (HR) and corresponding 95% confidence intervals (CI). In the univariate analysis, we included gender, age, smoking status, preoperative CHT, adjuvant CHT, thoracic RT, prophylactic cranial irradiation (PCI), stage, and type of operation. For multivariate survival analyses, the Cox regression model was adjusted for significant variables in the univariate analysis. Metric data are shown as median or mean and corresponding range or, in case of OS, as median and corresponding 95% CI. Clinical characteristics of patients with low PLR (LPLR) vs. (HPLR) or low NLR (LNLR) (vs. HNLR) were analyzed by Chi-square test in each individual cohort and then inter-cohort heterogeneity was assessed with the same test. Age (<65 years vs. ≥65 years) was considered as a categorical variable. To address the problem of multiple comparisons between clinicopathological parameters of the individual cohorts, Bonferroni's correction was applied. Thus, with six comparisons comprising three cohorts according to LNLR (vs. HNLR) or LPLR (vs. HPLR), $p$-values less than 0.008333 were considered to indicate statistical significance. All $p$-values were two-sided. Next, we evaluated the associations of preoperative NLR or PLR with clinicopathological characteristics (age, gender, stage, vascular involvement, tumor necrosis, peritumoral inflammation, and tumor grade) using Spearman's rank correlation which assess linear correlation coefficient as it measures the strength of linear relationship between the variables. The value of linear correlation coefficient ($r$) varies from −1 to 1 both values inclusive. No linear correlation ($r = 0$), weak positive linear correlation ($0.2 < r \leq 0.5$), moderate positive linear correlation ($0.5 < r \leq 0.8$), strong positive linear correlation ($0.8 < r < 1$), a $p$-value of less than 0.05 was considered to indicate statistical significance. All statistical analyses were performed using the PASW Statistics 22.0 package (SPSS Inc., Chicago, IL, USA).

## RESULTS

### Clinicopathological characteristics

Initially, we identified a total of 189 patients that underwent surgical resection: 91, 54, and 44 according to cohorts # 1, #2, and #3, respectively. PLR and NLR data were available in $n = 155$ cases (Fig. 1). Major clinicopathological characteristics of resected SCLC patients in pooled cohorts #1–3 ($n = 155$) are shown according to PLR and NLR in Table 1. The median age at diagnosis was 58 years. According to AJCC tumor staging, there were 60, 39, and 40 patients with Stage I, II, and III; respectively (16 patients had inaccurate staging data). During the follow-up period, 100 patients received adjuvant chemotherapy. The median preoperative neutrophil, lymphocyte, and thrombocyte counts were 4.621, 2.115, and 241.00; respectively, while NLR and PLR median values were 2.214 and 111.489. The identified cut-off PLR and NLR values for OS were 111.253 (sensitivity: 0.566, specificity: 0.589; Fig. S1A) and 2.258 (sensitivity: 0.545, specificity: 0.661; Fig. S1B), respectively. For DFS, PLR, and NLR value cut-offs of 112.174 (sensitivity: 0.531, specificity: 0.614; Fig. S1C) and 2.254 (sensitivity: 0.551, specificity: 0.649; Fig. S1D) were used, respectively.

Clinicopathological characteristics of resected SCLC patients in cohorts #1, #2, and #3 ($n = 189$) are shown according to NLR (Table S1) and PLR (Table S2). The 44 cases without NLR/PLR vs. 155 with NLR/PLR have no differences in clinicopathological

**Table 1** Major clinicopathological characteristics of resected SCLC patients according to neutrophil to lymphocyte ratio (NLR) and platelet to lymphocyte ratio (PLR) in the pooled cohorts #1–3 ($n = 155$).

| | | Total | | | | $p$ | Total | | | | $p$ |
|---|---|---|---|---|---|---|---|---|---|---|---|
| | | NLR | | | | | PLR | | | | |
| | | LNLR | | HNLR | | | LPLR | | HPLR | | |
| | | Count | #$n$ (%) | Count | #$n$ (%) | | Count | #$n$ (%) | Count | #$n$ (%) | |
| Gender | Male | 59 | (72) | 55 | (73.3) | 0.63 | 56 | (73.7) | 58 | (73.4) | 0.97 |
| | Female | 23 | (28) | 18 | (24.7) | | 20 | (26.3) | 21 | (26.6) | |
| Age | <65 | 63 | (76.8) | 45 | (61.6) | 0.04* | 58 | (76.3) | 50 | (63.3) | 0.06 |
| | ≥65 | 18 | (22) | 27 | (37) | | 18 | (23.7) | 27 | (34.2) | |
| | Unknown | 1 | (1.2) | 1 | (1.4) | | 0 | (0) | 2 | (2.5) | |
| Smoking | Non-smoker | 11 | (13.4) | 3 | (4.1) | 0.08 | 10 | (13.2) | 4 | (5.1) | 0.28 |
| | Smoker | 54 | (65.9) | 56 | (76.7) | | 54 | (71.1) | 56 | (70.9) | |
| | Ex-smoker | 6 | (7.3) | 3 | (4.1) | | 5 | (6.6) | 4 | (5.1) | |
| | Unknown | 11 | (13.4) | 11 | (15.1) | | 7 | (9.2) | 15 | (19) | |
| Preoperative CHT | No | 53 | (64.6) | 51 | (69.9) | 0.87 | 48 | (63.2) | 56 | (70.9) | 0.68 |
| | Yes | 21 | (25.6) | 19 | (26) | | 20 | (26.3) | 20 | (25.3) | |
| | Unknown | 8 | (9.8) | 3 | (4.1) | | 8 | (10.5) | 3 | (3.8) | |
| Adjuvant CHT | No | 17 | (20.7) | 14 | (19.2) | 0.64 | 10 | (13.2) | 21 | (26.6) | 0.054 |
| | Yes | 50 | (61) | 50 | (68.5) | | 52 | (68.4) | 48 | (60.8) | |
| | Unknown | 15 | (18.3) | 9 | (12.3) | | 14 | (18.4) | 10 | (12.7) | |
| Thoracic RT | No | 39 | (47.6) | 32 | (43.8) | 0.26 | 33 | (43.4) | 38 | (48.1) | 0.99 |
| | Yes | 18 | (22) | 23 | (31.5) | | 19 | (25) | 22 | (27.8) | |
| | Unknown | 25 | (30.5) | 18 | (24.7) | | 24 | (31.6) | 19 | (24.1) | |
| PCI | No | 45 | (54.9) | 44 | (60.3) | 0.9 | 40 | (52.6) | 49 | (62) | 0.53 |
| | Yes | 13 | (15.9) | 12 | (16.4) | | 13 | (17.1) | 12 | (15.2) | |
| | Unknown | 24 | (29.3) | 17 | (23.3) | | 23 | (30.3) | 18 | (22.8) | |
| Stage | I | 38 | (46.3) | 22 | (30.1) | 0.02* | 31 | (40.8) | 29 | (36.7) | 0.04* |
| | II | 17 | (20.7) | 12 | (16.4) | | 22 | (28.9) | 17 | (21.5) | |
| | III | 15 | (18.3) | 25 | (34.2) | | 12 | (15.8) | 28 | (35.4) | |
| | Unknown | 12 | (14.6) | 14 | (19.2) | | 11 | (14.5) | 5 | (6.3) | |
| pT | 1 | 21 | (25.6) | 17 | (23.3) | 0.73 | 18 | (23.7) | 20 | (25.3) | 0.42 |
| | 2 | 38 | (46.3) | 36 | (49.3) | | 37 | (48.7) | 37 | (46.8) | |
| | 3 | 10 | (12.2) | 12 | (16.4) | | 9 | (11.8) | 13 | (16.5) | |
| | 4 | 2 | (2.4) | 4 | (5.5) | | 1 | (1.3) | 5 | (6.3) | |
| | Unknown | 11 | (13.4) | 4 | (5.5) | | 11 | (14.5) | 4 | (5.1) | |
| pN | 0 | 42 | (51.2) | 34 | (46.6) | 0.33 | 40 | (52.6) | 36 | (45.6) | 0.13 |
| | 1 | 22 | (26.8) | 20 | (27.4) | | 21 | (27.6) | 21 | (26.6) | |
| | 2 | 10 | (12.2) | 19 | (26) | | 9 | (11.8) | 20 | (25.3) | |
| | Unknown | 8 | (9.8) | 0 | (0) | | 6 | (7.9) | 2 | (2.5) | |
| Vascular involvement | 0 | 32 | (39) | 26 | (35.6) | 0.17 | 34 | (44.7) | 24 | (30.4) | 0.29 |
| | 1 | 11 | (13.4) | 17 | (23.3) | | 13 | (17.1) | 15 | (19.1) | |
| | Unknown | 39 | (47.6) | 30 | (41.1) | | 29 | (38.2) | 40 | (50.6) | |

| | | Total | | | | p | Total | | | | p |
|---|---|---|---|---|---|---|---|---|---|---|---|
| | | NLR | | | | | PLR | | | | |
| | | LNLR | | HNLR | | | LPLR | | HPLR | | |
| | | Count | #n (%) | Count | #n (%) | | Count | #n (%) | Count | #n (%) | |
| Tumor necrosis | 0 | 18 | (22) | 16 | (21.9) | 0.36 | 15 | (19.7) | 19 | (24.1) | 0.6 |
| | 1 | 38 | (46.3) | 49 | (67.1) | | 43 | (56.6) | 44 | (55.7) | |
| | Unknown | 26 | (31.7) | 8 | (11) | | 18 | (23.7) | 16 | (20.3) | |
| Peritumoral inflammation | 0 | 20 | (24.4) | 11 | (15.1) | 0.22 | 20 | (26.3) | 11 | (13.9) | 0.4 |
| | 1 | 12 | (14.6) | 16 | (21.9) | | 12 | (15.8) | 3 | (3.8) | |
| | 2 | 8 | (9.8) | 13 | (17.8) | | 11 | (14.5) | 10 | (12.7) | |
| | 3 | 3 | (3.7) | 3 | (4.1) | | 4 | (5.3) | 2 | (2.5) | |
| | Unknown | 39 | (47.6) | 30 | (41.1) | | 29 | (38.8) | 53 | (67.1) | |
| Operation | Lob+seg | 67 | (81.7) | 57 | (78.1) | 0.57 | 66 | (86.8) | 58 | (73.4) | 0.52 |
| | PNO | 15 | (18.2) | 16 | (21.9) | | 10 | (13.2) | 21 | (26.6) | |
| | Unknown | 0 | (0) | 0 | (0) | | 0 | (0) | 0 | (0) | |

Notes:
HNLR, high neutrophil to lymphocyte ratio; LNLR, low neutrophil to lymphocyte ratio; HPLR, high platelet to lymphocyte ratio; LPLR, low platelet to lymphocyte ratio; $n$ %, percentage of total; CHT, chemotherapy; PCI, prophylactic cranial irradiation; pT, pathological T stage; pN, pathological N stage; RT, radiation therapy; Lob, lobectomy; Seg, segmentectomy; PNO, pneumonectomy; $p$, Chi-square (Fisher's exact) test.
# Data shown in parenthesis are column percentages.
* Statistically significant.

characteristics. Nevertheless, we found significant differences only in pathological stage according to PLR and NLR in the full cohort. A higher percentage of stage I patients had pre-treatment LNLR.

## Prognostic factors for OS and DFS

Univariate survival analysis identified significantly longer OS in patients with lobectomy and segmentectomy (vs. pneumonectomy, median OS, 56.2 vs. 31.7 months, $p = 0.015$, Fig. 2A), and also in patients with pathological N (pN) stage 0–1 (vs. pN 2, median OS, 64.9 vs. 30.3 months, $p = 0.001$; Fig. 2C). Moreover, OS was significantly longer in patients with LNLR (vs. HLNR, median OS, 74.8 vs. 44.5 months, respectively, $p = 0.003$, Fig. 3A), however there was no significant difference in OS according to PLR (LPLR vs. HPLR, median OS, 73.6 vs. 40.4 months, respectively, $p = 0.084$, Fig. 3C).

Univariate survival analysis also identified a significantly longer DFS in patients with lobectomy and segmentectomy (vs. pneumonectomy, median OS, 55.2 vs. 18.8 months, $p = 0.033$; Fig. 2B) and in patients with pN stage 0–1 (vs. pN 2, median OS, 55.7 vs. 23.0 months, $p = 0.002$; Fig. 2D), but there were no significant differences in DFS in patients with LNLR (vs. HNLR median OS, 68.8 vs. 34.9 months, $p = 0.051$, Fig. 3B) or LPLR (vs. HPLR, median OS, 68.8 vs. 32.8 months, $p = 0.086$, Fig. 3D).

The Cox multivariate analysis found that pN stage 0–1 (vs. pN 2, HR, 2.07; 95% CI [1.237–3.448]; $p = 0.006$) is a significant independent prognostic factor for OS, while the type of operation (lobectomy and segmentectomy vs. pneumonectomy, HR, 1.61; 95% CI [0.977–2.666]; $p = 0.062$) and LNLR (vs. HLNR, HR, 1.36; 95% CI [0.879–2.103];

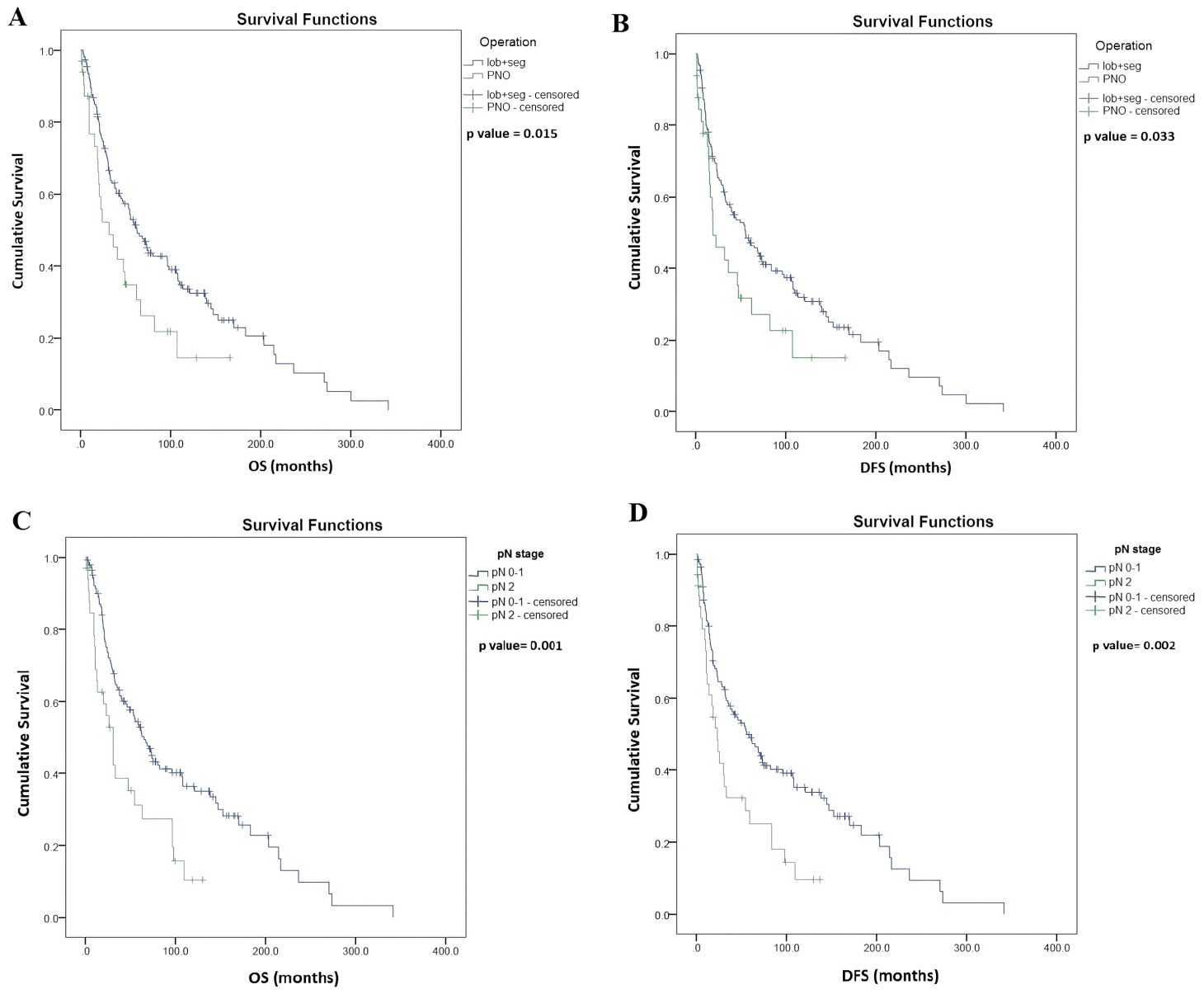

**Figure 2 Kaplan–Meier survival curves for overall survival (OS) and disease-free survival (DFS) in resected SCLC patients according to the type of lung resection surgery ($n = 187$) and lymph node status (LN) ($n = 176$).** (A) OS of patients underwent segmentectomy (seg) or lobectomy (lob) was significantly longer compared to those underwent pneumectomy (PNO) (median OS, 62.3 vs. 31.7 months, $p = 0.015$, log-rank test). (B) DFS of patients underwent segmentectomy or lobectomy (vs. pneumectomy; median OS, 55.2 vs. 18.8 months, $p = 0.033$, log-rank test). (C) OS of patients with pathological LN status (pN) 0–1 (vs. pN2, median OS, 64.9 vs. 30.3 months, $p = 0.001$, log-rank test. (D) DFS of patients with pN0–1 (vs. pN2, median OS, 55.7 vs. 23 months $p = 0.002$, log-rank test). Data on pN was not available in $n = 11$ cases as indicated in Table 1.              

$p = 0.167$) had no prognostic impact on OS. Furthermore pN stage 0–1 (vs. pN 2, HR, 2.041; 95% CI [1.241–3.357]; $p = 0.005$) had a prognostic impact also on DFS while the type of operation (lobectomy and segmentectomy vs. pneumonectomy, HR, 1.48; 95% CI [0.904–2.445]; $p = 0.119$), and LNLR (vs. HLNR, HR, 1.27; 95% CI [0.833–1.939]; $p = 0.226$) had no predictive value. Since the prognosis is significantly different for pN2
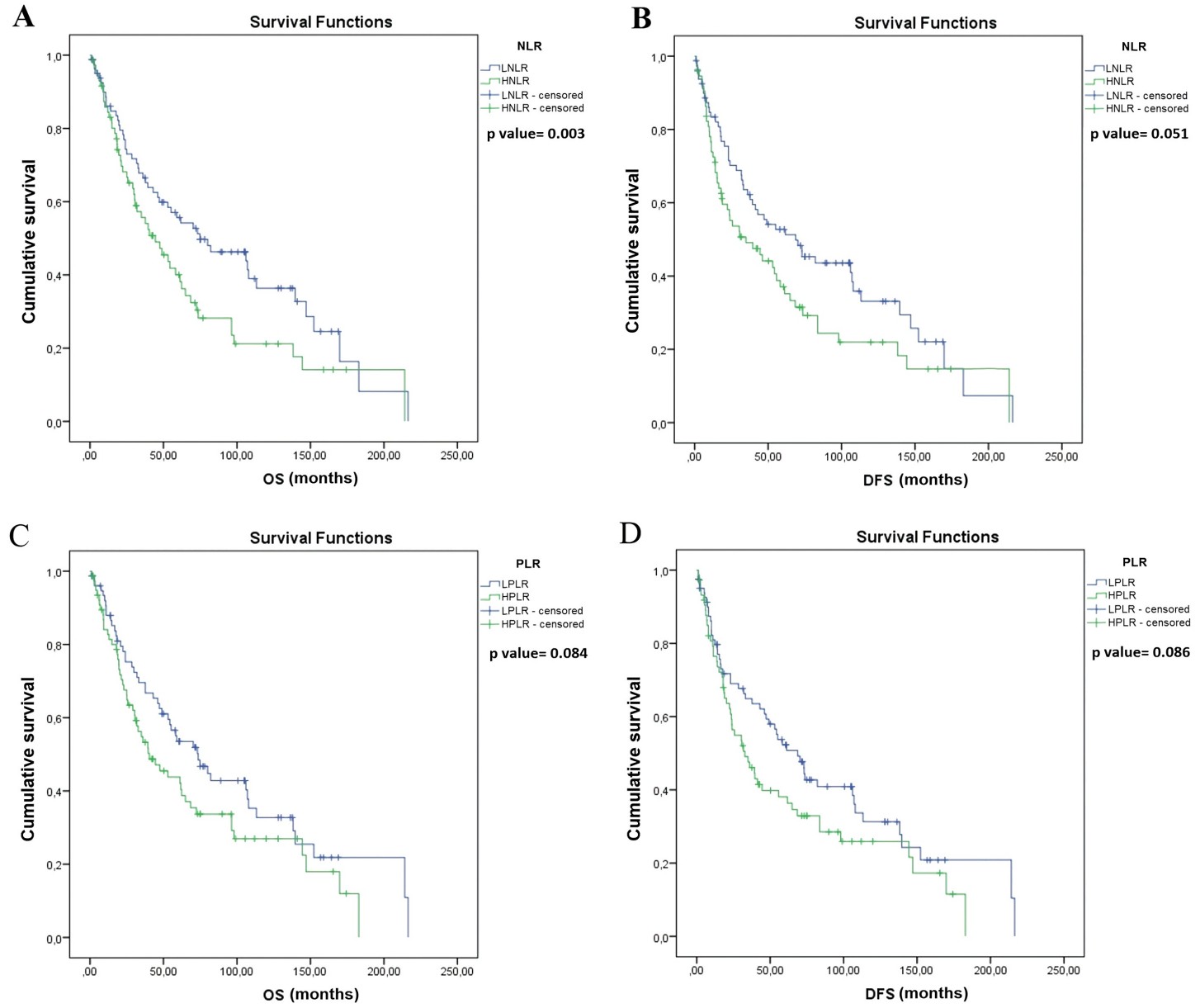

**Figure 3 Kaplan–Meier survival curves for overall survival (OS) and disease-free survival (DFS) in resected SCLC patients according to NLR and PLR ($n = 155$).** (A) OS of patients with low NLR (LNLR) (<2.25) was significantly longer compared to those with high NLR ((HNLR), median OS, 74.8 vs. 44.5 months, $p = 0.003$, log-rank test). (B) DFS of patients with LNLR (<2.25) (vs. HNLR median OS, 68.8 vs. 34.9 months, $p = 0.051$, log-rank test (not statistically significant)). (C) OS of patients with low PLR ((LPLR), <111) vs. (high PLR (HPLR), median OS, 73.6 vs. 40.4 months, $p = 0.084$, log-rank test (not statistically significant)). (D) DFS of patients with LPLR (<111) (vs. HPLR, median OS, 68.8 vs 32.8 months $p = 0.086$, log-rank test (not statistically significant)).

patients, we performed a univariate and a Cox multivariate analysis including only stage I and stage II patients, and we found that LNLR (vs. HNLR, HR, 1.582; 95% CI [1.010–2.478]; $p = 0.045$) is a significant prognostic factor for OS, as shown in Table 2. However, the type of operation (lobectomy and segmentectomy vs. pneumonectomy, HR, 1.697; 95% CI [0.974–2.958]; $p = 0.062$) had no prognostic value for stage I and stage II patients.

**Table 2 Clinical variables and survival of patients with stage I/II SCLC in the Cox proportional hazards model for overall survival.**

| | Univariate | | | Multivariate | | |
|---|---|---|---|---|---|---|
| Prognostic factor | HR | 95% CI | *p*-Value | HR | 95% CI | *p*-Value |
| NLR (LNLR vs. HNLR) | 1.621 | [1.036–2.537] | 0.035 | 1.582 | [1.010–2.478] | 0.045* |
| Type of operation (lobectomy and segmentectomy vs. pneumonectomy) | 1.780 | [1.061–2.986] | 0.029 | 1.697 | [0.974–2.958] | 0.062 |

**Notes:**
HR, hazard ratio; CI, confidence interval; HNLR, high neutrophil to lymphocyte ratio; LNLR, low neutrophil to lymphocyte ratio.
* Statistically significant.

The effects of adjuvant CHT on OS according to NLR and PLR are shown in Fig. S2. There were no significant differences in OS of patients with HNLR treated with adjuvant CHT compared to those did not receive adjuvant CHT (median OS, 53.2 vs. 40.4 months, respectively, *p* = 0.965, log-rank test, Fig. S2A). In contrast, OS of patients with LNLR received adjuvant CHT was significantly longer compared to those did not receive adjuvant CHT (median OS, 113.3 vs. 31.7 months, respectively, *p* = 0.013, log-rank test, Fig. S2B). We found no significant differences in OS of patients with HPLR that did not receive adjuvant CHT compared to those received adjuvant CHT (median OS, 25 vs. 61.3 months, respectively, *p* = 0.125, log-rank test, Fig. S2C). Moreover, there were no significant differences in OS of LPLR patients that did not receive adjuvant CHT compared to those treated with adjuvant CHT (median OS, 53.2 vs. 80 months, respectively, *p* = 0.481, log-rank test, Fig. S2D).

Next, we investigated the associations between preoperative NLR or PLR with clinicopathological characteristics and Spearman's correlation coefficient was calculated (Table 3). We found a statistically significant weak positive correlation between NLR and tumor stage ($r = 0.226$, $p = 0.007$), NLR and age ($r = 0.167$, $p = 0.038$), and NLR and PLR ($r = 0.035$, $p < 0.0001$). Of note, based on the limited case numbers in each individual cohort, we were not able to use ROC curves to identify individual cut-off values for combined analysis.

## DISCUSSION

Preoperative complete blood count (CBC) is often used as a broad screening test before lung cancer resection. CBC includes neutrophils, lymphocytes, and platelets, therefore NLR and PLR can easily be assessed in the routine clinical practice. NLR and PLR may reflect the extent of systemic inflammation elicited by cancer cells and have been proposed as reliable indicators of the host's inflammatory status. Lymphocytes play a key role in anti-tumor immune response by inhibiting tumor cell proliferation. HNLR indicates decreased lymphocyte count and/or increased neutrophil count. An increasing body of literature reports that tumor progression appears to be linked to the inflammatory response since multiple immune cells play critical roles in carcinogenesis (*Coffelt et al., 2010*; *DeNardo, Andreu & Coussens, 2010*; *Egeblad, Rasch & Weaver, 2010*; *Johansson et al., 2009*; *Pahler et al., 2008*). These studies reveal signaling molecules released by inflammatory cells that possess tumor-promoting features. These include the tumor growth factor EGF, the angiogenic growth factor VEGF, other proangiogenic factors such as Fibroblast Growth

| Table 3 Correlations of clinicopathological variables ($n = 155$). | | | |
|---|---|---|---|
| Factors | | Preop PLR | Preop NLR |
| Age | $r$ | 0.125 | 0.167 |
| | $p$ | 0.124 | 0.038* |
| | $N$ | 153 | 153 |
| Gender | $r$ | 0.003 | −0.038 |
| | $p$ | 0.970 | 0.635 |
| | $N$ | 155 | 155 |
| Smoking | $r$ | 0.089 | 0.088 |
| | $p$ | 0.306 | 0.314 |
| | $N$ | 133 | 133 |
| Vascular involvement | $r$ | 0.115 | 0.149 |
| | $p$ | 0.293 | 0.171 |
| | $N$ | 86 | 86 |
| Tumor necrosis | $r$ | −0.048 | 0.084 |
| | $p$ | 0.603 | 0.363 |
| | $N$ | 121 | 121 |
| Peritumoral inflammation | $r$ | 0.074 | 0.191 |
| | $p$ | 0.500 | 0.078 |
| | $N$ | 86 | 86 |
| pN | $r$ | 0.139 | 0.135 |
| | $p$ | 0.092 | 0.103 |
| | $N$ | 147 | 147 |
| pT | $r$ | 0.074 | 0.084 |
| | $p$ | 0.388 | 0.322 |
| | $N$ | 140 | 140 |
| Stage | $r$ | 0.158 | 0.226 |
| | $p$ | 0.064 | 0.007* |
| | $N$ | 139 | 139 |
| Preop NLR | $r$ | 0.357 | 1.000 |
| | $p$ | <0.001* | – |
| | $N$ | 155 | 155 |

Notes:
* Indicates significant correlation with Spearman test (mean OS cut).
NLR, neutrophil to lymphocyte ratio; PLR, platelet to lymphocyte ratio; $r$, correlation coefficient; $p$, probability value; $N$, number of patients.

Factor 2, chemokines, and cytokines that amplify the inflammatory state. Additionally, they may produce proangiogenic and/or proinvasive matrix-degrading enzymes, including Matrix Metallo Protease-9 and other matrix metalloproteinases, cysteine cathepsin proteases, and heparanase (*Qian & Pollard, 2010*; *Pahler et al., 2008*). Consistent with their expression of these diverse effectors, tumor-infiltrating inflammatory cells have been shown to induce and help sustain tumor angiogenesis, stimulate cancer cell proliferation, facilitate, via their presence at the margins of tumors, tissue invasion, and support the metastatic dissemination and seeding of cancer cells (*Coffelt et al., 2010*;

*Egeblad, Rasch & Weaver, 2010*; *Pahler et al., 2008*; *Qian & Pollard, 2010*; *Mantovani et al., 2008*; *Biswas & Mantovani, 2010*; *Joyce & Pollard, 2009*; *Wang et al., 2016*).

Previous studies investigated the role of PLR and NLR in other cancers. NLR in colorectal, gastroesophageal, pancreatic, ovarian, hepatocellular, lung, and renal cancer appears to be prognostic in patients with advanced-stage disease and can predict treatment response, which might be helpful in selecting patients for further therapy (*Guthrie et al., 2013*). Moreover, PLR was also reported to have prognostic value for poor prognosis in many cancers including ovarian cancer, gastric cancer, colorectal cancer, hepatocellular carcinoma, and NSCLC (*Zhou et al., 2014*).

In summary, in our surgically resected SCLC cohort, we found that LNLR was associated with significantly longer OS, while PLR had no prognostic impact. In line with our study, *Takahashi et al. (2015, 2016)* reported that East-Asian patient's preoperative HNLR is a significant predictor of poor prognosis in patients with completely resected stage I–III lung adenocarcinoma and stage I NSCLC. Moreover, according to two meta-analyses, pretreatment HPLR and HNLR is associated with poor OS, progression-free survival, and DFS in NSCLC patients, further supporting the predictive value of the previously mentioned biomarkers (*Qiang et al., 2016*; *Gu et al., 2016*).

The prognostic role of PLR might differ in NSCLC compared to SCLC might be explained by the different biological behavior, immunogenicity and molecular structure of these tumor types. In line with our study, in histologically combined-SCLC with NSCLC, *Shao & Cai (2015)* and *Wang, Jiang & Li (2014)* also showed that HNLR was associated with poor prognosis and recurrence. *Hong et al. (2015)* found that HNLR appeared to be associated with worse prognosis; however, this study could only show in univariate, but not in multivariate analysis.

In SCLC, to our knowledge, there are no available reliable cut-off values reported for NLR and PLR, and previous studies reported only combined limited- and extensive-stage SCLC patient cohorts predominantly from East Asia. Additionally, surgically resected SCLC patients have different prognosis compared to non-surgically treated limited- and extensive-stage cases (*Pan et al., 2017*; *Hamilton, Rath & Ulsperger, 2016*). Therefore, we used ROC analysis to determine the optimal cut-off values in our resected SCLC patients. In our study multivariate (and univariate) analysis identified LNLR (vs. HNLR) as a significant independent prognostic factor for OS. This may be particularly relevant considering that the main known prognostic marker is the mediastinal lymph-node involvement in limited-stage SCLC, and no clear prognostic marker is available to further individuate different prognostic groups.

Others investigated the role of PLR and NLR in large studies with patients of East Asian origin with limited- and extensive-stage SCLC patients and reported NLR and PLR cut-off values ranging from 3.2 to 5 and 122 to 258 (*Xie et al., 2015*).

A large study of 320 patients used the Cox proportional hazard model that showed that NLR $\geq 2.65$ (HR = 1.35; 95% CI [1.02–1.79]; $p = 0.039$), LDH $\geq 210$ (HR = 1.46; 95% CI [1.10–1.96]; $p = 0.002$), patients with surgery (HR = 0.55; 95% CI [0.33–0.93]; $p = 0.025$), thoracic RT (HR = 0.66; 95% CI [0.50–0.88]; $p = 0.005$) and PCI (HR = 0.71; 95% CI [0.53–0.96]; $p = 0.023$) were independent prognostic factors for OS in SCLC

patients (*Xie et al., 2015*). In our study, we had no data on LDH and we did not include PCI or thoracic RT in our analysis as an initial prognostic factor since patients received these modalities only at disease recurrence. In our study the cut-off value with ROC analysis was similar (NLR, 2.25). The difference can be explained by the early stage, resected patient cohort in our study, compared to a predominantly extensive-stage SCLC cohorts in other studies. However, the underlying pathological mechanism is similar.

Our results are in line with a recent study which showed that NLR but not PLR have prognostic value in SCLC (*Wang et al., 2016*; *Kang et al., 2014*). However, these studies included both extensive- and limited-stage disease.

In contrast another study showed that pretreatment NLR might be an independent and significant prognostic factor for OS only in extensive-stage SCLC while PLR in limited-stage SCLC (*Xie et al., 2015*). The difference can be explained that the latter study included pathologically (likely cytologically) diagnosed possibly unrevealed or unreported combined SCLC cases treated predominantly with CHT and RT compared to our homogenous surgically resected patients with histologically confirmed SCLC cohort.

To further highlight the clinical importance of these biomarkers another study group, which also found that increased levels of NLR and PLR were associated with worse clinical outcome, established and validated a novel nomogram for the prediction of OS in predominantly extensive-stage SCLC patients (*Pan et al., 2017*).

Nevertheless, previous SCLC studies included high number of patients with cases mainly on conventional treatment and East Asian origin, the difference to our study might be related to the fact that we included more homogenous, histologically confirmed and surgically resected Caucasian patients.

Moreover, in our study, higher PLR only showed a very weak, but significant correlation with higher NLR value. Also, higher NLR indicated a higher grade in tumor stage independent of LN status. However, in contrast to our SCLC study, in other types of lung cancer, NLR was associated with pathological factors such as tumor invasiveness, lymph node metastasis, poor differentiation, and vascular invasion (*Yang et al., 2016*). The difference can be explained by the different tumor behavior or immunogenicity.

The key limitations to our study are the retrospective nature and SCLC patients were pooled from three different institutions. Consequently, no information was available on the exact drugs, dose and cycles of the chemotherapy administered; therefore, the study cohort was relatively heterogeneous in terms of adjuvant chemotherapy and radiotherapy. Another limitation of the study was that there were no reported cut-off values for PLR and NLR in SCLC, hence the cut-off values were arbitrary obtained via ROC analysis.

## CONCLUSIONS

To our knowledge, this is the first study in Caucasian patients with resected SCLC which shows that LNLR (<2.25) with blood collected up to 30 days before surgical resection, may be a favorable prognostic factor for longer OS. The determination of LNLR should be further evaluated in other series of surgical resected patients and should be evaluated when planning trials addressed to define the optimal multimodal treatment in stage I–II SCLC. We also conclude that accurate mediastinal lymph node staging and NLR

and PLR may help in selecting patients for surgery in the future. Further prospective studies are needed to confirm these observations.

## ACKNOWLEDGEMENTS

The authors thank the patients and clinical teams.

### Funding

Zoltan Lohinai was supported by the ESMO Translational Research Fellowship, the 2018 LCFA-BMS/IASLC Young Investigator Scholarship Award, and the Hungarian Scientific Research Fund (OTKA #124652 and OTKA #129664). Zsolt Megyesfalvi and Balazs Santa received support from the Hungarian Scientific Research project (EFOP-3.6.3-VEKOP-16-2017-00009). The funders had no role in study design, data collection and analysis, decision to publish, or preparation of the manuscript.

### Grant Disclosures

The following grant information was disclosed by the authors:
ESMO Translational Research Fellowship.
The 2018 LCFA-BMS/IASLC Young Investigator Scholarship Award.
Hungarian Scientific Research Fund: OTKA #124652 and OTKA #129664.
Hungarian Scientific Research project: EFOP-3.6.3-VEKOP-16-2017-00009.

### Competing Interests

Glen J. Weiss receives personal fees from Paradigm, Angiex, IBEX Medical Analytics, Pfizer, IDEA Pharma, GLG Council, Guidepoint Global, Ignyta, Circulogene-all outside this work; has received travel reimbursement from Cambridge HealthTech Institute and Tesaro; ownership interest in Circulogene-outside this work; and has a patent for methods and kits to predict prognostic and therapeutic outcome in small cell lung cancer issued, outside this work. Other authors declare no potential conflicts of interest.

### Author Contributions

- Zoltan Lohinai conceived and designed the experiments, performed the experiments, analyzed the data, contributed reagents/materials/analysis tools, prepared figures and/or tables, authored or reviewed drafts of the paper, approved the final draft.
- Laura Bonanno conceived and designed the experiments, authored or reviewed drafts of the paper, approved the final draft.
- Aleksei Aksarin conceived and designed the experiments, authored or reviewed drafts of the paper, approved the final draft.
- Alberto Pavan conceived and designed the experiments, authored or reviewed drafts of the paper, approved the final draft.
- Zsolt Megyesfalvi conceived and designed the experiments, performed the experiments, contributed reagents/materials/analysis tools, authored or reviewed drafts of the paper, approved the final draft.

- Balazs Santa conceived and designed the experiments, performed the experiments, analyzed the data, contributed reagents/materials/analysis tools, authored or reviewed drafts of the paper, approved the final draft.
- Virag Hollosi conceived and designed the experiments, performed the experiments, authored or reviewed drafts of the paper, approved the final draft.
- Balazs Hegedus conceived and designed the experiments, authored or reviewed drafts of the paper, approved the final draft.
- Judit Moldvay conceived and designed the experiments, authored or reviewed drafts of the paper, approved the final draft.
- PierFranco Conte conceived and designed the experiments, authored or reviewed drafts of the paper, approved the final draft.
- Mikhail Ter-Ovanesov conceived and designed the experiments, authored or reviewed drafts of the paper, approved the final draft.
- Evgeniy Bilan conceived and designed the experiments, authored or reviewed drafts of the paper, approved the final draft.
- Balazs Dome conceived and designed the experiments, authored or reviewed drafts of the paper, approved the final draft.
- Glen J. Weiss conceived and designed the experiments, authored or reviewed drafts of the paper, approved the final draft.

### Human Ethics

The following information was supplied relating to ethical approvals (i.e., approving body and any reference numbers):

The study was conducted in accordance with the guidelines of the Helsinki Declaration of the World Medical Association and with the approval of the national level ethics committee (Hungarian Scientific and Research Ethics Committee of the Medical Research Council, ETTTUKEB-7214-1/2016/EKU), which waived the need for individual informed consent for this retrospective study. The Ethics Committee of Istituto Oncologico Veneto in Padova (Italy) evaluated and approved the study and patients provided informed consent to be signed for the collection, analysis and publication of data, when the investigators have the possibility to collect it from patients, according to Italian data protection authority dispositions. The ethics Committee of the Department of Health in Ugra (Russia) has evaluated and approved the study, which waived the need for individual informed consent for this retrospective study.

### Data Availability

The raw measurements are available in Dataset S1. The raw data shows the clinicopathological characteristics of the study population.

### Supplemental Information

Supplemental information for this article can be found online at http://dx.doi.org/10.7717/peerj.7232#supplemental-information.

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
