# Peer review of "Neutrophil–lymphocyte ratio is prognostic in early stage resected small-cell lung cancer"

_PeerJ, doi:10.7717/peerj.7232_

## Round 0.1 · original submission · Major Revisions

Based on the comments of all the reviewers, this manuscript needs a major revision. Please address all the questions and correct any syntax/grammatical errors.

Reviewer 1 ·

Basic reporting

No comment

Experimental design

No comment

Validity of the findings

No comment

Additional comments

This manuscript needs substantial revision.
Figure shown in manuscript is contradictory to the interpretation of the result.
This manuscript is very badly written which makes it very difficult to understand.
Figure used in this manuscript lacks figure legend.
Data doesn't support the finding of this manuscript.

·

Basic reporting

The manuscript entitled “Neutrophil-Lymphocyte ratio is prognostic in early-stage resected small-cell lung cancer” is a poorly-written manuscript. Overuse of abbreviations made this article difficult for reading. I strongly recommend authors to take help of professional English Language editors. This manuscript is a retrospective study aimed to investigate the prognostic value of neutrophil to lymphocytes ratio (NLR) and platelet to lymphocyte ratio (PLR) in Caucasian patients with resected small cell lung cancer (SCLC). Several studies already reported the prognostic value of NLR & PLR in SCLC. This study has a very narrow scope and may be interesting for oncologist who are mainly interested in SCLC.

Experimental design

No comment

Validity of the findings

Need further supportive study to confirm these results. But this study is well designed and statistically validated.

Additional comments

Major
1) Inclusion of the flow diagram of the study will be useful.
2) Research question was not clearly established. Background also need to be better written
3) Some part of the introduction is irrelevant especially Lines 63-67.
4) Poorly-written manuscript. Overuse of abbreviations made this article difficult for reading. I strongly recommend authors to take help of professional English Language editors.
5) Introduction and discussion look similar with same references which looks unnecessary.

Reviewer 3 ·

Basic reporting

English grammar /synatx used across the manuscript is acceptable. Minor grammatical errors have to be fixed in introduction section.

A lot of abbreviations do not have full forms in the text : for example
-Mention LNLR full form in abstract/background :
-What is ES and LS ?? early and late stage?
-CHT RT full forms should be stated in text

Authors have provided sufficient background in the introduction. However the second half of the introduction requires major revisions in terms of flow. The literature about the discrpencies in the levels of NLR and PLR and patient prognosis has to be explained better. Right now it is messy and reader can get lost! Authors should also clearly discuss the possible reasons that could result in these opposite conclusions for all the studies. For example different race or study design or treatment regimen etc..

Authors have to clearly state how similar or different their analysis is from existing data. Authors do specify that this cohort is caucasian compared to Asian cohorts used in other studies cited. Why do they think that this analysis will be useful/conclusive?

Ethics statement is provided and raw data is shared.

Figures require better annotation. X axis unit is not mentioned. P value should be stated on charts.

In Table 1
-Only gender has total but no other category?
-What is pT, pN in table1?
-“Column N%” should be relabeled for more clarity!


Other minor grammatical errors :
"Consecutive patients" with histologically confirmed and surgically resected SCLC evaluated between 2000 and 2013 at three centers were retrospectively analyzed. NLR and PLR at diagnosis was used to categorize patients into "high" (H) and "low" (L) groups based on ROC analysis
In a large cohort of 2,476 "SCLC patients underwent" surgical resection and chemotherapy had improved five-year overall survival (OS) compared to those who received nonsurgical therapy alone

Experimental design

Research question is defined but how this research will fill a knowledge gap is not clearly stated. This can be elaborated further.

The individual cohort sizes were not very big in the study so the authors combined three cohorts to get about 155 patients for subsequent analysis. While this number is not very big it is still a reasonable size for this kind of study.

Authors should comment on any issues faced when using data from multiple institutes. For example data from all three cohorts were measured/graded the same or comparable way across different centers (For example NLR and PLR calculations, histological grade etc) ?

Statistical analysis described in Methods section seem appropriate for the study. Why was p value less than 0.008333 used for significance?Authors should state what was the p value used for Spearman correlation analysis.

Authors have to provide more details for each statical analysis in the results section. Details are very messy and not clearly explained.

Authors should clearly state how NLR and PLR were calculated? Were they log transformed for symmetric distribution?

Authors should clearly state in results section (before Table 1!) how NLR and PLR were classified as Low or High. Authors mention in Methods section that they use the ROC curve (instead of using a cutoff). This analysis has to be further explained clearly in the results with the appropriate figures. this section is fundamental for the entire paper and is not present in the Results section.

Where is the data/figure for lines 187-190 of results?

In describing table 1 there is no mention of “vascular involvement, tumor necrosis, peritumoral inflammation, or tumor grade” in the results text..All results stated in figures/tables/ should be discussed and vice versa.

In this paper all chemotherapy treated patients across different institutes were grouped together for all analysis. If the data is available, drugs administered to patients should be specified. If patients were grouped by specific drugs administered would there be differences between groups?

The results section under Prognostic factors for OS and DFS is extremely unclear. All the data mentioned in the text is not supported by the figure. For example authors state the following :
"Univariate survival analysis identified significantly longer OS in patients with lobectomy and segmentectomy (vs. pneumonectomy, median OS, 56.2 vs 31.7 months, p=0.0015), pN 0-1 (vs 195 pN 2, median OS, 64.9 vs 30.3 months, p=0.001), and LNLR (vs HLNR, median OS, 74.8 vs 196 44.5 months, p=0.033, Fig 1A). " However Fig 1A only has survival curve for LNLR vs HLNR survival.

Authors should clearly state the individual variables used for the multivariate analysis.

Authors state that "Thus higher NLR indicates a higher grade in tumor stage. Also, higher PLR correlates with a higher NLR value" While the p values are significant the r values are seem very low to make these statements? (r=0.23 or r=0.035?)

In Table 3 was analysis done for 189 patients (as stated in table title)? Authors state NLR/PLR data only available for 155 patients in methods section?

Validity of the findings

Please see section above for details to be verified to ensure the data and analysis is robust.

Discussion and Conclusion is reasonable but requires better flow and organization (in discussion section)

Additional comments

This study has the potential of being useful to the scientific community because of the valuable patient information for caucasians with SCLC. However in its current state the manuscript is very difficult to read and comprehend. While the aim of the study is clear and the introduction is reasonably explained, the results sections require major revision! The results have to be stated more clearly and in an organized manner. Appropriate results have to be clearly tabulated, figures have to be shown for all data reported in the text, all data in the tables have to be mentioned in the results. While the statistical approaches described in the Method section seem appropriate for the study, the details are not clearly mentioned in the Results section for each analysis.

Reviewer 4 ·

Basic reporting

This study attempts to demonstrate prognostic significance of preoperative NLR and PLR in patients with resected SCLC. The finding of this study is novel, but there are several issues in this study as follows. In summary, they need to revise this manuscript correctly.

Experimental design

1) In the Introduction section, rationale of the study is unclear, particularly regarding prognostic significance of preoperative NLR in patients with resected NSCLC. Preoperative high NLR was reported to be significantly associated with poorer survival specifically in NSCLC (Please refer to “Takahashi Y et al. Ann Surg Oncol 2015;22 Suppl 3:S1324-31.” and “Takahashi Y et al. World J Surg 2016: ;40(2):365-72.”). Please do not ignore important literature for this topic.
2) The authors should describe the inclusion and exclusion criteria for patient selection.
3) The study cohort should be homogeneous especially in terms of treatment protocol. However, the study cohort seems to be very heterogeneous in terms of adjuvant chemotherapy and radiotherapy. At least we would know whether NLR is a prognostic factor in each patient group (adjuvant therapy + or -).
4) Also, patients with stage III SCLC should be excluded in the survival analysis because they have a very poor prognosis. Or, they may show survival curves of each stage.
5) Are there any differences in NLR and PLR between the institutions. The analytical method can be different in each institution.
6) In Figure 1, what does the X-axis represent? They should put unit in the X-axis.

Validity of the findings

7) In the Discussion, biological background of correlation between NLR and inflammation as well as immuno-microenvironment seems to be insufficient. The careful discussion should improve quality of their novel finding.
8) The Cut-off value may be arbitrary unless it is referred from previous study. Thus, they may add this description as one of the limitation of their study.

---

## Round 0.2 · accepted · Accept

Manuscript is ready for publication. Authors have addressed all the questions raised by reviewers.

Reviewer 1 ·

Basic reporting

No comment

Experimental design

No comment

Validity of the findings

No comment

Additional comments

This manuscript provides an interesting data to consider neutrophil to lymphocyte ration as a prognostic marker for multimodal treatment for SCLC.
The authors have clarified the issues raised in previous review. However, discussion part is very lenghty which needs to be polished and crisped.
Line 246-255 is not required for the context of current study.